# Evaluation of the Hypotensive Preterm Infant: Evidence-Based Practice at the Bedside?

**DOI:** 10.3390/children10030519

**Published:** 2023-03-06

**Authors:** Elizabeth Murphy, David B. Healy, Roberto Chioma, Eugene M. Dempsey

**Affiliations:** 1Department of Paediatrics & Child Health, University College Cork, T12 YT20 Cork, Ireland; 2INFANT Research Centre, University College Cork, T12 DC4A Cork, Ireland; 3Department of Life Sciences and Public Health, Fondazione Policlinico Universitario Agostino Gemelli IRCCS, 00168 Rome, Italy

**Keywords:** very preterm neonates, hypotension, NIRS, non-invasive cardiac output monitoring

## Abstract

Choosing the appropriate management approach for the preterm infant with low blood pressure during the transition period generally involved intervening when the blood pressure drifted below a certain threshold. It is now clear that this approach is too simplistic and does not address the underlying physiology. In this chapter, we explore the many monitoring tools available for evaluation of the hypotensive preterm and assess the evidence base supporting or refuting their use. The key challenge relates to incorporating these outputs with the clinical status of the patient and choosing the appropriate management strategy.

## 1. Introduction

For far too long, the management of babies with low blood pressure has been overly standardised; typically, if the mean blood pressure fell below a certain threshold, a series of interventions were executed, the goal of which was to improve the mean blood pressure value, often irrespective of the clinical condition of the infant [1,2]. Evidence from randomised controlled trials of cardiovascular support suggests that such an approach results in an increase in mean blood pressure in over two-thirds of cases [3,4,5]. This basic approach was the mainstay of management for over 40 years and, in many ways, its ‘effectiveness’ has contributed to the current paucity of evidence available to guide cardiovascular support in the preterm population. We now understand that in the first days of adaptation, this simplistic management approach may be inappropriate and potentially deleterious [6,7]. Numerous trials comparing different inotropic agents in a ‘head-to-head’ comparison evaluated short-term outcomes [4,8,9,10,11,12,13,14]. Despite being carried out with great rigor, the inclusion criteria and the definition of the underlying ‘hypotension’ varied. Thus, one of the key questions remains unanswered: how should one define cardiovascular compromise in the very preterm infant in the first days of life?

This complexity has driven the development of technologies and strategies aiming to manage this common clinical scenario more precisely. In this review, we will explore the many monitoring tools available for evaluation of the hypotensive preterm. We will assess the current evidence base supporting their use in the preterm population. Whilst these devices may provide very valuable information, the onus of decision making still rests on the bedside clinician. The key challenges relate to making ‘sense’ of the information provided by objective bedside monitoring tools, whether they be continuous or intermittent, and incorporating these findings with the clinical status of the patient at any particular time. Thus, appreciating the shortcomings of clinical evaluation is important, but its inclusion remains critical.

## 2. Bedside Clinical Assessment

Bedside clinical assessment typically includes an evaluation of the newborn’s colour, capillary refill time, urine output, and overall activity level. Not only are these prone to significant variability but, independently, their relationship with cardiovascular instability in preterm infants is also ill-defined. Colour has been shown to be very subjective and prone to significant interrater variability. O’Donnell and colleagues highlighted the marked discrepancy between skin colour and oxygen saturation when neonatologists reviewed video recordings from the delivery suite [15]. Attempts have been made to objectify colour assessment [16,17]. DeFelice et al. found that skin colorimeter values differed significantly between high- and low-illness-severity groups, suggesting that it may have a role in evaluation [16]. However, no data on blood pressure or cardiac function were provided. Skin perfusion has also been investigated as a clinical marker of circulatory status. Osborn et al. reported a weak association between capillary refill time and superior vena cava flow [18]. Wodey et al. identified a significant association between cardiac index on echocardiography and capillary refill time in preterm neonates [19]. Previously, we identified a poor relationship between capillary refill values obtained in different body locations and simultaneously obtained superior vena cava (SVC) flow measurements. Peripheral capillary refill time has marked variability (up to 10 s in healthy term infants), and therefore, we would generally advocate the use of central measurements [20].

Bedside measures of lactate and acid–base status are helpful but, again, need to be interpreted carefully secondary to the normal physiological changes that occur in the first days of a preterm infant’s life. Extremely preterm infants experience a normal self-limiting, non-anion gap metabolic acidosis with a peak typically on day four of life. High lactate values at three hours of life have been associated with adverse short-term outcomes in extremely preterm infants [21]. Similarly, Phillips found that the initial lactate value was significantly higher in very low birth weight infants who died and was highest in the first 12 h (10.2 vs. 3.84 mmol/L, *p* < 0.0001). A lactate value of ≥6.9 mmol/L has a sensitivity of 77% and specificity of 78% to predict mortality [22]. Sequential lactate values provide more information about the overall status than isolated values [23,24].

Preterm infants have a characteristic pattern of urinary output in the first week of life [25] characterised by a low urinary output initially followed by a marked diuresis secondary to a rise in atrial natriuretic peptide [26]. The presence of adequate urine output on day one provides reassurance, but its evolution in the subsequent 24–48 h is equally important. The same can be said of all these individual parameters, but when combined, they allow for better identification of potentially low-flow states. A prolonged central capillary refill time (>4 s) combined with a lactate value greater than 4 mmol/L was found to have a specificity for identifying low superior vena cava flow of 97% [27]. These data are consistent with other studies documenting improved detection rates of low-flow states by including more than one clinical or laboratory parameter [28]. The adoption of a relevantly holistic approach incorporating clinical, biochemical, and standard haemodynamic parameters may accurately identify infants in or at risk of low-flow circulatory states.

Objective measures readily available at the bedside include heart rate and blood pressure values. Less commonly considered is the pulsatility index. Measures often performed in paediatric and adult intensive care such as central venous pressure measurements are not practical in the care of very preterm infants. Where it has been performed, normal values for CVP in preterm infants have a wide range (2.8–13.9 mmHg), and most infants who have cardiovascular instability during the first days are typically not volume-depleted, thus limiting their clinical utility. Heart rate varies with gestational and postnatal age and can be influenced by many factors, such as stress, temperature, pain, and medication administration. As such, absolute heart rate values may poorly reflect systemic perfusion. Despite its shortfalls, valuable information may also be obtained by interrogation of its evolution over time, and it is an important parameter to always consider.

The primary driver of intervention remains the mean blood pressure (BP) value, measured either invasively or noninvasively. Noninvasive BP measurements can be challenging [29]. The Haemodynamics Working Group of the International Neonatal Committee published a review addressing methods of BP measurement in neonates. This review highlighted some of the key challenges in obtaining reliable noninvasive BP measurements and lists recommendations for standardising these measurements [30]. Ideally, in a preterm infant at risk of cardiovascular instability, invasive BP monitoring is recommended.

Numerous BP reference ranges exist, a comparison of which reveals notable differences. Ranges are often based on gestational age, birth weight, and postnatal age criteria [31,32,33,34]. These statistically determined values vary considerably, highlighting the significant methodologic differences in how they were obtained. Some advocate for single absolute mean BP values chosen over a wide range of gestational ages and a wide timespan. Examples include an absolute mean BP less than 30 mmHg or mean blood pressure less than the gestational age equivalent measured in mmHg (e.g., 25 weeks’ gestation, mean BP value of 25 mmHg). The evidence base supporting any of these recommendations is weak. Miall-Allen et al. found that a mean BP of less than 30 mmHg for over an hour was *associated with* severe intracranial haemorrhage, ischemic cerebral lesions, or death within 48 h in a cohort of 33 infants of less than 31 weeks’ gestation [35]. No severe cerebral lesions developed in infants with a mean BP of 30 mmHg or greater. This served as evidence to maintain a mean BP greater than 30 mmHg. Advocating this approach means that almost all infants at 23 weeks and some at 24 weeks will require inotrope administration, which certainly does not fit with an individualised approach to the management of the extremely preterm infant. Likewise, the British Association of Perinatal Medicine rule is not evidence-based but remains the commonest criterion to intervene [36]. It is a simple rule to remember but completely neglects the normal dynamic evolution of BP over the first day of life. Its use beyond the first days of life is illogical [37,38].

The different components of the BP measurement may give us greater insight into the underlying circulatory status. The systolic component largely reflects left ventricular contractility. Low systolic pressures, therefore, may be caused by a decreased preload, an increase in left ventricular afterload, or a reduction in myocardial contractility. Diastolic blood pressure reflects the resting vascular tone and the intravascular blood volume and is important in determining coronary blood flow. Causes of diastolic hypotension include reduced preload and/or decreased systemic vascular resistance. Thus, considering these individual components may provide better insight into the underlying physiology.

Whilst awareness of the limitations of BP monitoring is essential, it remains an important biomarker of circulatory status. One should be cognisant of the natural increase in BP in the first days and that BP does not equate directly with flow. It should be incorporated into decision-making algorithms but rarely be the sole criterion to initiate therapy. Perhaps it is best to think of blood pressure values in terms of evaluation pressures and trigger pressures. Figure 1 represents a pragmatic approach to patients with cardiovascular instability as evidenced by a low mean blood pressure value. The evaluation pressures are values that warrant further interventions such as echocardiography, and the trigger pressures may be cumulative evidence such as duration of low blood pressure and other measures that may warrant intervention. We conclude that the mean BP at or below gestational age equivalent may serve as a useful evaluation pressure in the first day of life.

Unlike assessing a child with neonatal encephalopathy and deciding on instituting therapeutic hypothermia, no such validated scoring system or assessment tool is available to guide interventions in preterm infants at risk of cardiovascular instability, a condition much more readily encountered in the neonatal unit. What is clear is that there is significant value in including clinical, biochemical, and objective bedside data prior to deciding on management approaches. Perhaps one of the greatest additions to this assessment over the last 10 years has been the incorporation of echocardiography to enhance decision making.

## 3. Echocardiography

Echocardiography use in the neonatal intensive care unit (NICU) has grown exponentially over the last decade as evidenced by an increase in publications relating to its use, the growth of numerous targeted neonatal echocardiography (TNE) programmes across the world [39], and the recently published international guidelines developed in this area of newborn care. TNE is not designed to replace the expertise of paediatric cardiology; in trained hands, it can add real-time, accurate haemodynamic information to aid clinical management. A number of expert consensus statements encourage the development of a framework for the implementation of standard echocardiography assessment and reinforce the necessity of structured training [40,41], and its inclusion in neonatal training programs has been advocated [42,43]. Given the dynamic nature of cardiac ultrasound, and considering the technical challenges in premature newborns, there is potential for great inter-observer variability and inaccuracy. These documents advise a structured approach to echocardiography to address specific indications. One potential concern relates to missing structural lesions; however, some retrospective data showed that the concordance of diagnostic echocardiography is high in centres with established training programmes [44]. However, access to paediatric cardiology review remains essential.

Over the last 10 years, guidance on neonatal haemodynamic assessment with TNE has evolved but is primarily by consensus and experience [40]. While research is ongoing on preterm circulatory (patho)physiology, studies in this area remain challenging due to issues around patient selection, recruitment, and inter-operator variability. To date, the evaluation of its role in guiding management has primarily been observational. Nevertheless, TNE is rational and objective and can provide valuable information on circulatory status when performed by skilled operators. Several recent publications have documented the incorporation of TNE into the haemodynamic assessment of preterm infants [45,46,47,48] with resultant changes in clinical management following assessment. These have highlighted the indications for its use, including the assessment of patent ductus arteriosus, persistent pulmonary hypertension of the newborn, and, more recently, systolic hypotension. However, a direct change in management does not necessarily imply improved clinical outcomes, which have yet to be demonstrated. A national survey on circulatory management in Japan reported that most neonatologists perform functional echocardiography in extremely preterm infants and do so frequently in the first days of life. The use of vasodilators is high in this population of infants [49]. However, King et al. recently reported on the variation in the use of echocardiography in North American centres [50]. Variation was greatest in the first few days of life and was associated with increased exposure to nitric oxide and vasoactive medications. The increasing use of very early echocardiography was not associated with the increasing treatment of the PDA. However, confounding by indication cannot be accounted for in this study design. This retrospective study highlights the variability of echo use and the uncertain impact of echocardiography use on clinical outcomes. Thus, it is critical that centres incorporating echocardiography establish protocols, review processes, and governance structures to ensure that changes that impact clinical management are robustly reviewed and evaluated.

The exact role of echocardiography in the evaluation of the hypotensive preterm infant has yet to be delineated. Measurements such as volume status and right and left ventricular output may not be as useful in this population. There is no agreed objective measure of volume status in the preterm infant. TNE measurements commonly used for this aim include inferior vena cava diameter. Markers of the inferior vena cava diameter are often of limited use in this population given their inaccuracy with positive pressure ventilation. Left ventricular end-diastolic diameter is influenced by variations in preload and afterload. Measures of cardiac output are prone to shunt-related effects, and determining normative ranges is problematic. More advanced tissue doppler imaging may overcome some of these challenges. While TNE may provide valuable information regarding cardiac dysfunction or other abnormalities, in the setting of transitional hypotension with normal clinical parameters, a normal TNE assessment should provide reassurance that non-intervention is appropriate, thus avoiding potentially harmful medications.

Thus, several questions remain to be answered. Should all preterm infants of less than 28 weeks’ gestation have an early echo performed to identify potential cardiovascular compromise? This would be very resource-heavy and would result in a significant number of infants having an echocardiogram performed who ultimately do not require the handling involved in performing an echocardiogram. Should only those infants deemed to be at risk of cardiovascular compromise have an echocardiogram performed? They may be difficult to identify but may include those not in receipt of antenatal corticosteroids or delayed cord clamping, those mechanically ventilated, and perhaps the more extreme preterm infants such as those born below a certain gestational age, e.g., below 25 weeks. Should only those infants who demonstrate potential cardiovascular compromise have an echocardiogram performed, such as those with a borderline mean blood pressure so-called ‘evaluating blood pressure’? These questions remain to be answered. Ultimately, the information provided by echocardiography may help to better understand the underlying problem but may not be sufficient to accurately identify low-flow states in such a complex transitional circulation. Other technologies are required to augment our assessment of end-organ blood flow. These are outlined in Figure 2 and discussed in more detail in the following sections.

## 4. Cerebral and Somatic NIRS

Indirect, continuous, objective, and noninvasive monitoring of end-organ perfusion is possible using near-infrared spectroscopy (NIRS). Commercially available neonatal NIRS sensors have been developed with the primary focus of cerebral monitoring, and several centres have highlighted the potential short- and long-term benefits of cerebral NIRS monitoring [51,52,53,54,55]. Several groups continue to evaluate the role of splanchnic (ArSO_2_), renal, and peripheral muscle oximetry as measured by NIRS to improve outcomes relating to cardiovascular dysfunction, PDA, and necrotising enterocolitis (NEC). While the technology is safe and has been adopted into routine clinical practice in many centres, the evidence to support resultant improvement in outcomes is lacking. The evaluation of ArSO_2_ as an assessment tool for necrotising enterocolitis (NEC) is ongoing, but, as NEC aetiology is considered multifactorial and not solely rooted in low-blood-flow states, its discussion is outside the scope of this review. Furthermore, no work has been conducted to simultaneously assess splanchnic blood flow and ArSO_2_ [56]. Hypotensive VLBW infants (BP < 10th centile) have been demonstrated to have reduced muscle oxygen delivery and oxygen consumption compared to normotensive infants [57]. This normalised with the treatment of hypotension. However, a more recent study implied that this relationship was independent of blood pressure alone and was, instead, more related to cardiac function as estimated by fractional shortening [58].

Studies that have evaluated cerebral and peripheral NIRS concurrently note differences in readings presumably because of adaptive cerebral blood flow due to cerebral autoregulation (CAR). However, in the context of hypotension, very premature infants are at greater risk of cerebral injury given their immature CAR pathways. It is presumed that in such circumstances, failure of autoregulatory capacity would be reflected by falling CrSO_2_. Nevertheless, a few studies have demonstrated that CrSO_2_ values are not significantly different between preterm infants with and without mild to moderate hypotension [52,59]. Pichler et al. found that it was not possible to reduce the burden of hypotension in moderately preterm infants using management strategies guided by cerebral NIRS [60]. The above results may be compounded by inaccurate clinical definitions of hypotension and, moreover, individual CAR tolerances for blood pressure variation [61]. We previously demonstrated that the duration of cerebral hypoxia was longer in hypotensive compared to non-hypotensive infants [62]. However, among treated hypotensive infants, CrSO_2_ was no different in those treated with dopamine compared to placebo, indicating that simply improving mean arterial pressure may not mitigate the adverse outcome. Transfer function gain, a derived parameter from the CrSO_2_ signal, may be a better marker of impaired autoregulation in hypotensive infants than the absolute values [63]. Ultimately, in the management of hypotensive infants, cerebral oximetry may have utility for the identification of autoregulatory failure and, in combination with clinical signs and other objective parameters, guide patient selection with regard to therapeutic intervention [64]. The term ‘integrated evaluation of haemodynamics’ has been coined. This is, in essence, the incorporation of cerebral NIRS measurements and echocardiography parameters in clinical decision-making pathways [65,66]. The authors describe a shorter recovery time during the new integrated evaluation of haemodynamics in a pre–post retrospective study [66].

## 5. Noninvasive Cardiac Output Monitoring

Cardiac output (CO) is a key physiological parameter to assess the haemodynamic status of transitioning infants, being one of the major determinants of arterial BP in combination with systemic vascular resistance. A complex interplay exists between these forces to determine systemic blood flow and end-organ perfusion. Low- and high-CO states have been associated with increased morbidity and mortality and adverse neurodevelopmental and respiratory outcomes [67,68]. As the gold standard CO measurement uses invasive techniques, such as thermodilution via pulmonary artery catheterisation [69], it is not feasible in infants due to technical limitations and risk of morbidity [70]. As the cardiovascular physiology of preterm neonates is dynamic and echocardiography provides information for a certain time period, noninvasive cardiac output monitoring (NICOM) technologies have been investigated in this population. These techniques offer continuous noninvasive measurements of CO via various electrical biosensing technologies [71], are easy to use and apply, and are recordable alongside other physiological monitors [72,73]. Electrical biosensing technology encompasses two broad categories: bioimpedance (BI) and bioreactance (BR). BI derives CO from the change in impedance (electrical conduction) produced by the degree of erythrocyte alignment in the aorta throughout the cardiac cycle. BR is based on the theory that blood flow changes not only produce variations in impedance but also in capacitance and inductance (the ability of biological tissue to store energy in electrical and non-electrical forms, respectively). BR derives an estimate of CO by measuring the phase shift of an oscillating current as it traverses the thorax. Significant technical and practical differences exist between BI and BR, such as underlying algorithms, electrode dimensions and placement, susceptibility to environmental interference, and confounding factors on accuracy [71,73,74,75,76,77]. Due to these aspects, BI and BR should not be considered interchangeable. A significant research effort has been invested over the last decade into the assessment of NICOM accuracy as compared to echocardiography-derived estimates of CO, showing conflicting results [78,79]. According to current evidence, interpreting NICOM values with thresholds based on echocardiography values is problematic [71,72]. NICOM has been employed in several observational studies to monitor stable preterm infants during the first days of life [76,80], unveiling some insights into the physiology of the haemodynamic transition in this population. Miletin et al. recently published BR-derived values in a small group of more immature preterm infants between 6 and 48 h of life [81]. In accordance with previous reports [80], they observed lower CO at 6 h, with a subsequent increase over the second day of life. BP followed a similar trend, with a weak, although significant, correlation with CO (*r* = 0.71, *p* < 0.001). Interestingly, infants developing intraventricular haemorrhage and/or necrotising enterocolitis showed initial lower CO followed by a more pronounced increase, suggesting ischemia-reperfusion injury as a pathophysiological mechanism. Furthermore, the lowest registered CO was lower in infants with adverse outcomes, while the lowest mean BP did not differ significantly between the groups. Hence, NICOM seems to provide a more sensitive marker of haemodynamic imbalance in transitioning preterm infants compared to the traditional monitoring of BP. However, the study was conducted on a small population with few adverse outcomes, and a formal analysis of the predictive value of CO in the case of hypotension was not performed. Despite these encouraging reports, concerns remain regarding NICOM accuracy in particular conditions commonly encountered in sick preterm infants during the transition, such as invasive and noninvasive ventilation [77,79,82], the persistence of foetal shunts [77,83], and low- and high-CO states [77]. Hence, more research is required before introducing NICOM in clinical practice. It appears crucial to perform validation studies employing a true gold standard reference method, develop reference values and clinical decision limits for the specific NICOM technologies, and, finally, assess their usefulness in clinical trials. We currently do not advocate the use of noninvasive cardiac output monitoring to direct clinical decision making in the preterm infant.

## 6. EEG

An alternate method of assessment of end-organ blood flow is to measure brain activity with EEG or aEEG. A normal EEG in the setting of low blood pressure or cardiac output may suggest that cerebral autoregulation is intact. However, this is something not routinely utilised in preterm infants in this setting, and EEG assessment is also very much gestation-dependent. However, a few groups have explored the relationship between measures of brain activity and cardiac output/blood pressure. Weindling et al., in an observational study of 35 very low birth weight infants, found that cerebral activity was abnormal when the mean blood pressure was less than 23 mmHg, but the overall number with abnormal EEG was low [84]. In a similar study, they found no relationship between left or right ventricular output (RVO) and EEG power measures. They noted that if the cardiac output was low but blood pressure was normal (>30 mmHg), then the EEG was normal, suggesting intact cerebral autoregulation [85]. Shah et al. performed an observational study evaluating the relationship between aEEG measures and measures of cardiovascular status in a population of 90 extremely low gestational age newborns. Those neonates treated with inotropes at 12 h for clinical management of low BP or poor perfusion had persistently low aEEG amplitude and continuity in the first 48 h of life [86]. Pereira et al. found no difference in the maximum and minimum aEEG amplitude between the three intervention thresholds utilised in this study [87]. This suggests a poor relationship between blood pressure values and brain activity levels although it is acknowledged by the authors that periods when blood pressure may have been lowest were missed. These studies highlight that there is potential for the inclusion of EEG as a measure of cardiovascular well-being in preterm infants. However, many significant practical challenges remain before EEG could be seen as a useful monitoring tool in the setting of cardiovascular instability in the ELBW infant.

## 7. Multimodal Monitoring in Cardiovascular Assessment

An enormous amount of continuous physiological data are generated daily for an individual NICU patient. This extraordinarily rich data source is currently underappreciated and, in many centres, underutilised. Often, the data are captured in the patient electronic health records hourly, which significantly limits its potential usefulness. Real-time continuous monitoring of heart rate, blood pressure, the pulsatility index, oxygen saturation, carbon dioxide, cerebral and somatic oxygenation, and EEG and, ultimately, developing decision support tools are very appealing and would allow a more personalised approach to care. Understanding the complexity of cardiovascular physiology in the first days of life and trying to decipher the interplay between many of these parameters is key. Neonatal care has seen some recent advances in the diagnosis and management of certain conditions utilising real-time data acquisition. Heart rate variability has proved useful in detecting infants at risk of sepsis [88]. Seizure detection algorithms have been shown to reduce the overall seizure burden [89].

Efforts have been made to develop real-time haemodynamic monitoring systems but remain a long way off from incorporating them routinely into clinical care. With advances in biomedical engineering, the capability to collect, filter, store, and analyse huge quantities of physiological data is now feasible. Relating these to clinical outcomes, utilising artificial intelligence, and developing decision support tools are the next key steps. Some centres have developed such ‘monitoring towers’ to allow the capture of these data [90]. Others have developed bedside systems that allow real-time detection of cerebral autoregulation [63,91]. Developments such as these provide a foundation for future management of neonatal cardiovascular compromise.

## 8. Conclusions

The evaluation of cardiovascular compromise at the bedside remains a key challenge in neonatology. The goal is to identify infants at the earliest stage possible stage to initiate therapies to avoid unnecessary morbidity and, importantly, to prevent the unnecessary use of potentially deleterious interventions. In the early stages of cardiovascular compromise, clinical evaluation may be falsely reassuring, thus introducing the necessity to re-evaluate the infant frequently. Echocardiography can provide important objective measures of cardiovascular status and currently remains the most important objective monitoring tool currently available. Continuous bedside tools such as cerebral NIRS may be important. NICOM is very appealing but has many limitations at present. The development of multimodal monitoring platforms with incorporated decision support tools has the potential to transform the diagnosis and management of cardiovascular compromise. The next decade should see a paradigm shift in the evaluation of preterm cardiovascular instability.

## Figures and Tables

**Figure 1 children-10-00519-f001:**
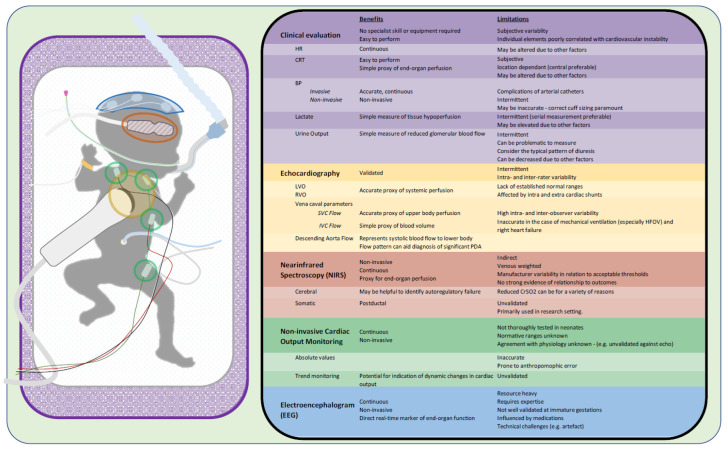
The benefits and limitations of clinical evaluation, echocardiography, NIRS, NICOM, and EEG in the assessment of preterm infants with potential cardiovascular instability.

**Figure 2 children-10-00519-f002:**
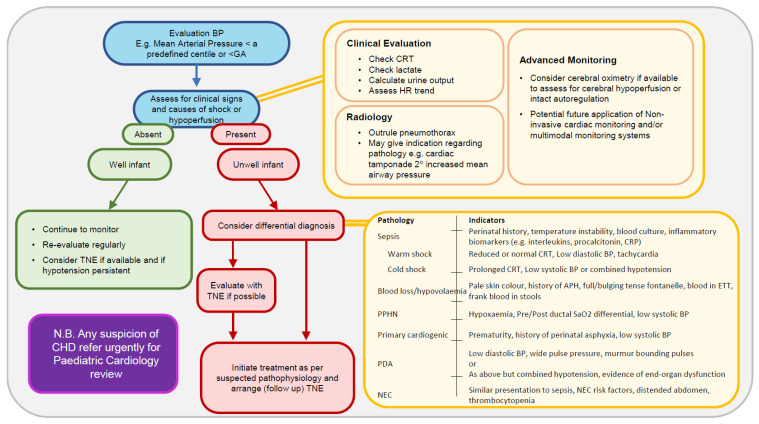
A pragmatic blood-pressure-based approach to the assessment of the preterm infant with potential cardiovascular instability.

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
