# Peer review of "Evaluation of the Hypotensive Preterm Infant: Evidence-Based Practice at the Bedside?"

_children, 2023, doi:10.3390/children10030519_

Round 1

Reviewer 1 Report

Excellent review of a pervasive problem in neonatology. I agree with the approach, the validation of the positions, and the hypotheses regarding adjunct diagnostic tools. However, I'm not very convinced about the role of gestational age as an indicator of mean blood pressure without additional supporting documentation.

Author Response

Response:

Thank you for your review of the manuscript.

Blood pressure in preterm infants is subject to a high degree of individual patient variation, with gradual elevation of mean arterial pressure (MAP) over time. MAP is straightforward to obtain via an arterial catheter and is accurate and continuous. We are in agreement with the reviewer and, nor are we convinced about the role of the “gestational age rule”, or any other arbitrarily defined cut off for that matter. Indeed, for many very preterm infants with MAP below their gestational age, the “hypotension” is merely transient, does not result in end-organ hypoperfusion, and requires no intervention. Nevertheless, MAP is a familiar parameter on the basis of which a more detailed assessment can be instigated. If an institution already employs a non-individualised management strategy (e.g. BP below a particular centile, or BP below the GA rule) we suggest the alternative approach of using a threshold for evaluation rather than for treatment. This assessment cue should encourage the clinician to attempt to determine whether, in any given infant, their “hypotension” represents a low-flow state and if end-organ hypoperfusion is occurring, and ultimately better inform whether treatment, if any, is required. The reason for including this is based on the results of previous surveys and recommendations. The data from the German Neonatal Network presents data that is very consistent with low blood pressure on day one of life. Please see Table below form this data set (Faust et al. 2015, PMID: 26199082, DOI: 10.1136/archdischild-2014-306483 ).

Reviewer 2 Report

This review is an interesting summarization of the current knowledge on the evaluation studies for the hypotension in preterm infants, which can help guide the afterward interventions and personalized management strategies. The authors did a great job in recapping the collected information and streamlining the covered tests for hypotensive preterm infants in the figures as a useful workflow. Overall, the manuscript and figures are logically organized, and the topic brings some scientific importance to this work. I listed several concerns need to be addressed.

1.   There are many English language errors and improper use in the text, including but not limited to: line 20 “interventions occurred” to “interventions were executed”; line 33 “strategies aimed at” to “strategies aiming at”; line 37 “decision making rests on” to “decision making relies on”; line 195 “in in the evaluation” two ins; line 219 “ultimately” to “At last”; line 376 “and as importantly to” to “and importantly, to”.

Please do extensive polishing to the language and correct the errors.

2.   The information in this manuscript is already largely summarized in previous publications (ie, PMID: 31236020) and guidelines. What is the major novelty in this manuscript?

3.   For the readers’ benefit and interest, it is suggested to discuss more about the chemical tests for biomarkers in preterm infants hypertension, such as inflammatory factors IL-6 and IL-33 (ie, PMID: 33407770, 30207217, 23906495), which were not fully summarized before.

Author Response

Reviewer 2

This review is an interesting summarization of the current knowledge on the evaluation studies for the hypotension in preterm infants, which can help guide the afterward interventions and personalized management strategies. The authors did a great job in recapping the collected information and streamlining the covered tests for hypotensive preterm infants in the figures as a useful workflow. Overall, the manuscript and figures are logically organized, and the topic brings some scientific importance to this work. I listed several concerns need to be addressed.

Response:

Thank you for your review of our manuscript. We appreciate your time and comments.

  1. There are many English language errors and improper use in the text, including but not limited to: line 20 “interventions occurred” to “interventions were executed”; line 33 “strategies aimed at” to “strategies aiming at”; line 37 “decision making rests on” to “decision making relies on”; line 195 “in in the evaluation” two ins; line 219 “ultimately” to “At last”; line 376 “and as importantly to” to “and importantly, to”.

Please do extensive polishing to the language and correct the errors.

Response:

We have corrected these and any other English language errors.

  1. The information in this manuscript is already largely summarized in previous publications (ie, PMID: 31236020) and guidelines. What is the major novelty in this manuscript?

Response:

We believe that many guidelines for hypotension in very preterm neonates emphasise the use novel technologies and that valuable information that can be gained by a primary clinical assessment is overlooked. While we agree that these are useful modalities for patient assessment, appropriate context of the clinical picture needs to be provided to enable their correct interpretation. Furthermore, some technologies such as NIRS and NICOM should still be considered under development and caution should be employed when using these clinically. Any manuscript discussing implementation of such technologies should discuss these limitations, which we believe is an important addition. We have also provided information to the reader to make their own judgements on the potential role of these tools. Additionally, we acknowledge that not all institutions have the luxury of 24/7 access to modalities such as echo or non-invasive monitors, or the skilled personnel they require.

As such, we have structured our manuscript and algorithm in such a way to illustrate that an assessment of a hypotensive infant can be performed without immediate access to these tools, allowing treatment decisions in the short-term with appropriately scheduled investigations, such as echo, thereafter, or facilitating informed discussion with and referral to a higher-level centre if necessary.

  1. For the readers’ benefit and interest, it is suggested to discuss more about the chemical tests for biomarkers in preterm infants hypertension, such as inflammatory factors IL-6 and IL-33 (ie, PMID: 33407770, 30207217, 23906495), which were not fully summarized before.

Response:

We wished to focus on cardiovascular instability, not on the issue of neonatal sepsis. There are limited biomarkers available in the assessment of cardiovascular instability, and none that we are aware of that is sensitive and specific enough. We have addressed two blood biomarkers, lactic acid and base deficit. However on the basis of your suggestion we have added potential markers of sepsis in Figure 2.

Round 2

Reviewer 2 Report

This review is an interesting summarization of the current knowledge on the evaluation studies for the hypotension in preterm infants, which can guide the following interventions and personalized management strategies. The authors did a great job in recapping the collected information and streamlining the covered tests for hypotensive preterm infants in the figures as a useful workflow. Overall, the manuscript and figures are logically organized, and the topic brings some scientific value to this work. The authors responded to my questions and made revisions. It is suggested to enlarge the font in figure 1 which is hard to read.

Author Response

Thank you for further reviewing our manuscript.

In response to your comments we have enlarged the font in Figure 1 and also delineated rows within the table of Figure 1 with subtle colour differences as can be seen in the attached file.
